# Task Arithmetic Through The Lens Of
# One-Shot Federated Learning

**Zhixu Silvia Tao**[*]                                    *zhixu.tao@princeton.edu*
*Operations Research and Financial Engineering*
*Princeton University*

**Ian Mason**                                              *imason@fujitsu.com*
*Fujitsu Research of America*

**Sanjeev Kulkarni**                                       *kulkarni@princeton.edu*
*Operations Research and Financial Engineering*
*Electrical and Computer Engineering*
*Princeton University*

**Xavier Boix**                                            *xboix@fujitsu.com*
*Fujitsu Research of America*

**Reviewed on OpenReview:** *https://openreview.net/forum?id=Cgyo7S7Oy0*

## Abstract

Task Arithmetic is a model merging technique that enables the combination of multiple models' capabilities into a single model through simple arithmetic in the parameter space, without the need for additional fine-tuning or access to the original training data. However, the factors that determine the success of Task Arithmetic remain unclear. In this paper, we examine Task Arithmetic for multi-task learning by framing it as a one-shot Federated Learning problem. We demonstrate that Task Arithmetic is mathematically equivalent to the commonly used algorithm in Federated Learning, called Federated Averaging (FedAvg). By leveraging well-established theoretical results from FedAvg, we identify two key factors that impact the performance of Task Arithmetic: data heterogeneity and training heterogeneity. To mitigate these challenges, we adapt several algorithms from Federated Learning to improve the effectiveness of Task Arithmetic. Our experiments demonstrate that applying these algorithms can often significantly boost performance of the merged model compared to the original Task Arithmetic approach. This work bridges Task Arithmetic and Federated Learning, offering new theoretical perspectives on Task Arithmetic and improved practical methodologies for model merging. Our code is available at: https://github.com/SilviaTao/one_shot_fl_task_arithmetic

## 1 Introduction

With the proliferation of fine-tuned models across diverse domains, efficiently combining these models to achieve excellence across multiple tasks has emerged as a critical research challenge. *Task Arithmetic* (Ilharco et al., 2023), a recent technique in model merging, offers a simple yet effective solution. Consider a set of $T$ tasks, each defined by a training dataset and a loss function for fine-tuning. Let $\theta_{\text{pre}} \in \mathbb{R}^d$ be the parameters of a pre-trained model and $\theta_t \in \mathbb{R}^d$ be the fine-tuned model parameters on task $t$. A *task vector* $\tau_t$ is generated by subtracting the pre-trained model parameters from the fine-tuned model parameters: $\tau_t = \theta_{\text{pre}} - \theta_t$. The sum of these task vectors, $\sum_{t=1}^{T} \tau_t$, produces a direction that enhances performance of

---

[*]Work was done while Zhixu was an intern at Fujitsu Research of America.

the pre-trained model across multiple tasks for which the fine-tuned models have been trained. In other words, updating the pre-trained model parameters by $\theta_{\text{pre}} + \sum_{t=1}^{T} \tau_t$ results in a multi-task model. A key advantage of this approach is that it only involves element-wise operations in the weight space, eliminating the need for additional fine-tuning.

Despite its strong empirical performance, Task Arithmetic lacks substantial theoretical understanding. Only a small number of works have investigated this empirical success theoretically (Ortiz-Jimenez et al., 2023; Porrello et al., 2024). In this paper, we take a step towards bridging the gap between theory and practice by framing Task Arithmetic as a form of one-shot Federated Learning.

Federated Learning (McMahan et al., 2017), a distributed machine learning paradigm, enables devices to collaboratively train one shared model without exchanging the raw data. Federated Learning's goal is to retain data privacy and reduce computational costs, as all raw data remains stored locally on edge devices. In a typical Federated Learning training process, a server coordinates the training process by iterating through the following steps (Kairouz et al., 2021). First, the server broadcasts the current global model parameters and a training program to all the devices. Then each device locally computes an update to the model by using its own data. Finally, the server aggregates all the local updates from devices and updates the current global model by using the aggregated local updates. A commonly used algorithm for this training process is Federated Averaging (FedAvg) (McMahan et al., 2017). In one-shot Federated Learning, the server learns a global model in only a single round of communication between itself and all the devices (Guha et al., 2019).

In this work, we show that using one-shot FedAvg is equivalent to Task Arithmetic, thus offering a new perspective on Task Arithmetic through the lens of one-shot Federated Learning. Using the connection between Federated Learning and Task Arithmetic, we can leverage the extensive theoretical and algorithmic advancements in Federated Learning to better understand when Task Arithmetic is effective and how it can be improved. To the best of our knowledge, this is the first study to bridge Federated Learning and Task Arithmetic. Our main contributions and the structure of the paper are summarized as follows.

**Bridge Task Arithmetic and Federated Learning:** In Section 3, we establish the connection between Task Arithmetic and one-shot Federated Averaging, formalizing Task Arithmetic by using notions from Federated Learning.

**Analyze the Impact of Data and Training Heterogeneity in Task Arithmetic:** In Section 4, we use existing theoretical results from Federated Learning to deepen our understanding of Task Arithmetic. Specifically, we analyze how data heterogeneity, which slows convergence in FedAvg, also slows the convergence of Task Arithmetic (Section 4.1), and how training heterogeneity, which causes objective inconsistencies in Federated Learning, similarly impacts Task Arithmetic (Section 4.2).

**Identify and Adapt Federated Learning Algorithms for Task Arithmetic:** In Section 5, we adapt several Federated Learning algorithms to mitigate the challenges posed by data and training heterogeneity. To achieve effective adaptions, we first discuss challenges in adapting Federated Learning algorithms (Section 5.1), and then we recommend four Federated Learning algorithms to enhance Task Arithmetic (Section 5.2).

**Experimentally Show That Federated Learning Algorithms Often Improve Task Arithmetic:** We conduct experiments on vision-language model CLIP (Radford et al., 2021) for image classification tasks in Section 6, and on large language model Llama2 (Touvron et al., 2023) for instruction following, mathematical reasoning and code generation in Section 7. Our experiments confirm that adapting Federated Learning algorithms often improves the merged model's performance compared to Task Arithmetic.

## 2   Related Work

Our work bridges Federated Learning and Task Arithmetic, a prominent approach within the growing domain of model merging. Thus, this section reviews related work on both model merging and Federated Learning.

**Model Merging**   Task Arithmetic is one of many recent works on model merging (Wortsman et al., 2022a; Frankle et al., 2020b; Wortsman et al., 2022b; Matena & Raffel, 2022; Ainsworth et al., 2022; Don-Yehiya et al., 2022). Though the term model merging is relatively new, firstly formalized by Matena & Raffel (2022), the concept has received significant investigation (Wortsman et al., 2022b;a; Izmailov et al., 2018; Yang et al.,

2019; Tarvainen & Valpola, 2017; Gupta et al., 2020). For example, Wortsman et al. (2022b) average the pre-trained model parameters and fine-tuned parameters to enhance the robustness of fine-tuned model against distribution shifts. Wortsman et al. (2022a) show that averaging parameters of multiple fine-tuned models with different hyperparameter configurations can improve robustness and accuracy. Izmailov et al. (2018) demonstrate that averaging parameters along a single SGD trajectory leads to better generalization.

Task Arithmetic, introduced by Ilharco et al. (2023), refines model merging by introducing task vectors and a hyperparameter $\lambda$ to control how much task vectors modify pre-trained model parameters. This method has inspired various follow-up work on using simple arithmetic operations for model merging (Yu et al., 2024; Zhang et al., 2023; Ortiz-Jimenez et al., 2023; Stoica et al., 2023; Yadav et al., 2024a; Yang et al., 2023) such as sparsifying task vectors (Yu et al., 2024), merging parameter-efficient modules (Zhang et al., 2023), fine-tuning in linearized model spaces (Ortiz-Jimenez et al., 2023) and resolving sign interference of task vectors (Yadav et al., 2024a).

A concurrent work of Task Arithmetic is Jin et al. (2022), who propose the *RegMean*. When merging two linear regression models, the model merging problem can be formulated as an optimization problem which has a closed-form solution. RegMean uses this solution to merge every linear layer of fine-tuned models. Unlike Task Arithmetic, RegMean is limited to linear layers and requires access to the input features of all linear layers. While Jin et al. (2022) also note that model merging is an extreme case of Federated Learning with single communication round, they do not explore this further. Moreover, RegMean focuses on merging linear layers, distinguishing it from our framework which addresses more general model merging problems.

There is a substantial body of research dedicated to understanding the effectiveness of model merging (Draxler et al., 2018; Entezari et al., 2021; Frankle et al., 2020a; Garipov et al., 2018; Benton et al., 2021; Li et al., 2018; Foret et al., 2020; Simsek et al., 2021). Some studies focus on the theory of linear model connectivity (Draxler et al., 2018; Entezari et al., 2021; Frankle et al., 2020a; Garipov et al., 2018), while others emphasize the flatness of the loss landscape (Benton et al., 2021; Li et al., 2018; Foret et al., 2020; Simsek et al., 2021). However, there has been relatively little work addressing the effectiveness of Task Arithmetic except for Ortiz-Jimenez et al. (2023); Porrello et al. (2024). Our work aims to bridge this gap.

**Federated Learning**  In Federated Learning, *FedAvg* (McMahan et al., 2017) is widely used to solve the following distributed optimization problem across $M$ devices: $\min_{\theta \in \mathbb{R}^d} L(\theta) := \frac{1}{M} \sum_{m=1}^{M} L_m(\theta)$ where $L_m(\theta) := \mathbb{E}_{x_m \sim \mathcal{D}_m}[\ell(\theta, x_m)]$ is the objective function on each device $m$, defined by some loss function $\ell$ and data distribution $\mathcal{D}_m$. The core idea behind *FedAvg* is to perform local Stochastic Gradient Descent (local SGD) on each device, followed by model averaging on the server. A substantial body of research has analyzed the performance of FedAvg and local SGD (Li et al., 2019; Karimireddy et al., 2020; Khaled et al., 2020; Koloskova et al., 2020; Woodworth et al., 2020b;a; Glasgow et al., 2022; Patel et al., 2024; Wang et al., 2022; Woodworth et al., 2021).

A key challenge in the theoretical analysis of FedAvg arises from data heterogeneity, where each device $m$ has a different data distribution $\mathcal{D}_m$. In the homogeneous setting where $\mathcal{D}_m = \mathcal{D} \; \forall m$, Woodworth et al. (2021) have established the min-max complexity of FedAvg with smooth and convex loss functions. In the more complex heterogeneous setting, various works have derived the convergence rate of FedAvg under different assumptions about data heterogeneity (Li et al., 2019; Khaled et al., 2020; Koloskova et al., 2020; Woodworth et al., 2020b;a; Glasgow et al., 2022; Patel et al., 2024; Wang et al., 2022). In this work, rather than focusing on extending existing theoretical results, we focus on using these results to analyze Task Arithmetic.

To address the challenges posed by data heterogeneity, extensive research has focused on designing algorithms to improve the performance of FedAvg (Karimireddy et al., 2020; Reddi et al., 2020; Li et al., 2020b; Ye et al., 2023b; Li et al., 2021b; Tenison et al., 2022; Wang et al., 2020). Some work has enhanced optimization algorithms by regulating the local training process (Karimireddy et al., 2020; Reddi et al., 2020; Li et al., 2020b; Wang et al., 2020; Li et al., 2021b), while others have proposed alternative aggregation methods beyond simple averaging (Tenison et al., 2022; Ye et al., 2023b). Another line of work focuses on personalized Federated Learning (Smith et al., 2017; Tan et al., 2022; Mansour et al., 2020; Li et al., 2021a; Finn et al., 2017), addressing data heterogeneity by adapting the global model locally for each device.

Aside from data heterogeneity, Wang et al. (2020) notice different local training process (which we refer to training heterogeneity) exacerbates the convergence of federated optimization algorithms, leading them to converge to a stationary point of an objective function inconsistent with the original objective function.

## 3 Task Arithmetic is One-Shot FedAvg

To deepen our understanding of the mechanism behind Task Arithmetic in multi-task learning, in this section, we establish a connection between one-shot FedAvg and Task Arithmetic.

Given $T$ tasks, the objective in multi-task learning is to train a model parameterized by $\theta \in \mathbb{R}^d$ that performs well across all $T$ tasks. This can be formulated as minimizing the following multi-task objective function:

$$L(\theta) = \frac{1}{T} \sum_{t=1}^{T} L_t(\theta). \tag{1}$$

Here $L_t(\theta) = \mathbb{E}_{(x_t,y_t) \sim \mathcal{D}_t}[\ell(\theta; x_t, y_t)]$ represents the objective function for task $t$, where $(x_t, y_t)$ is a pair of input and output drawn from the data distribution $\mathcal{D}_t$, and $\ell(\cdot)$ denotes the loss function associated with the data and model. This formulation aligns with that used in Federated Learning, where each device $t$ has a local objective function $L_t$. $L(\theta)$ is referred to as the global objective function.

In Federated Learning, the global objective function (1) is often optimized using FedAvg. In FedAvg, each local objective function is optimized through several iterations of Stochastic Gradient Descent (SGD), after which the server averages all the local updates. This process, also known as *local Stochastic Gradient Descent* (local SGD), is repeated over multiple communication rounds. Formally, given $R$ communication rounds and initial global model parameters $\theta_0$, FedAvg follows the following update process $\forall r \in [R]$:

$$
\begin{aligned}
\theta_{t,r}^{(0)} &= \theta_{r-1} \quad \forall t \in [T] \\
\theta_{t,r}^{(k+1)} &= \theta_{t,r}^{(k)} - \eta_{t,r}^{(k)} g_t(\theta_{t,r}^{(k)}) \quad \forall k \in [0, K_t - 1], \forall t \in [T] \\
\theta_r &= \theta_{r-1} + \frac{\beta}{T} \sum_{t=1}^{T} (\theta_{t,r}^{(K_t)} - \theta_{r-1}) = \theta_{r-1} - \frac{\beta}{T} \sum_{t=1}^{T} \sum_{k=0}^{K_t-1} \eta_{t,r}^{(k)} g_t(\theta_{t,r}^{(k)}).
\end{aligned}
\tag{2}
$$

Here, $\theta_r$ denotes the global model parameters obtained by optimizing the global objective at the end of the $r$-th communication round, and $\theta_{t,r}^{(k)}$ denotes the local model parameters obtained by optimizing the $t$-th local objective at the $k$-th local step during round $r$. The learning rate used for this step is $\eta_{t,r}^{(k)}$. The stochastic gradient of the $t$-th local objective function $L_t$ is $g_t(\cdot)$, and $\beta$ is the outer step size used to aggregate all local updates. In the one-shot setting where $R = 1$, the update simplifies to the following:

$$\theta_{OS} = \theta_0 + \frac{\beta}{T} \sum_{t=1}^{T} (\theta_t^{(K_t)} - \theta_0) = \theta_0 - \frac{\beta}{T} \sum_{t=1}^{T} \sum_{k=0}^{K_t-1} \eta_t^{(k)} g_t(\theta_t^{(k)}) \tag{3}$$

where $\theta_{OS}$ denotes the parameters generated by one-shot FedAvg.

In Task Arithmetic, the procedure mirrors the process in FedAvg. Each task $t$ independently minimizes its own objective function $L_t$ by performing $K_t$ iterations of SGD with learning rates $\{\eta_t^{(0)}, \ldots, \eta_t^{(K_t-1)}\}$, starting from the same pre-trained model parameters $\theta_0$ and converging to a minimizer $\theta_t^* \in \arg \min L_t(\theta)$. This yields $\theta_t^* = \theta_t^{(0)} - \sum_{k=0}^{K_t-1} \eta_t^{(k)} g_t(\theta_t^{(k)})$ where $\theta_t^{(0)} = \theta_0 \; \forall t$. The task vector $\tau_t$ is defined by $\tau_t = \theta_t^* - \theta_t^{(0)}$. Using Task Arithmetic, a new set of parameters can be constructed as

$$\theta_{TA} = \theta_0 + \lambda \sum_{t=1}^{T} \tau_t = \theta_0 - \lambda \sum_{t=1}^{T} \sum_{k=0}^{K_t-1} \eta_t^{(k)} g_t(\theta_t^{(k)}) \tag{4}$$

where $\lambda$ is a hyperparameter called the scaling coefficient (Ilharco et al., 2023), which controls the extent to which the sum of task vectors is added back to the pre-trained parameters. By comparing equations (3) and (4), we see that performing Task Arithmetic is equivalent to one-shot FedAvg with outer step size $\beta = \lambda T$.

# 4 Adapting Federated Learning Theory for Task Arithmetic

In this section, we extend theoretical insights from Federated Learning to Task Arithmetic, identifying two main factors that impact its performance: *data heterogeneity* and *training heterogeneity*. We analyze how these factors impact the convergence of Task Arithmetic to the global minimum of a convex objective function and to the local minimum of a non-convex objective function.

## 4.1 Data Heterogeneity

This subsection is dedicated to understanding how data heterogeneity influences the performance of Task Arithmetic. Data heterogeneity is common in Federated Learning and refers to the situation when data on each device is non-independent and identically distributed (non-i.i.d.) (Li et al., 2020b; Ye et al., 2023a; Wen et al., 2023). In the context of Task Arithmetic, we define data heterogeneity as follows:

**Definition 4.1.** *(Data Heterogeneity) In Task Arithmetic, data heterogeneity refers to the non-i.i.d nature of training data across different tasks.*

In the convergence analysis of FedAvg, data heterogeneity has been a longstanding issue. Given the connection between FedAvg and Task Arithmetic, in order to understand how data heterogeneity impacts Task Arithmetic, it is helpful to first review existing findings on data heterogeneity in FedAvg. We begin by introducing several standard assumptions commonly used in the literature (Li et al., 2019; Karimireddy et al., 2020; Khaled et al., 2020; Koloskova et al., 2020; Woodworth et al., 2020b;a; Glasgow et al., 2022; Patel et al., 2024; Wang et al., 2022; Woodworth et al., 2021).

**Assumption 4.2.** *(Convexity and Smoothness) Assume all the task objective functions $L_t$ are convex and $H$-smooth. That is, $\forall t \in [T]$ and $\forall \theta, \varphi \in \mathbb{R}^d$, $L_t(\theta) \leq L_t(\varphi) + \langle \nabla L_t(\varphi), \theta - \varphi \rangle + \frac{H}{2} \|\theta - \varphi\|^2$.*

**Assumption 4.3.** *(Bounded Stochastic Noise) The stochastic gradient computed by each task is unbiased with bounded variance. That is, $\forall \theta \in \mathbb{R}^d$,*

$$\mathbb{E}_{(x_t, y_t) \sim \mathcal{D}_t}[\nabla \ell(\theta; x_t, y_t)] = \nabla L_t(\theta) \quad and \quad \mathbb{E}_{(x_t, y_t) \sim \mathcal{D}_t}[\|\nabla \ell(\theta; x_t, y_t) - \nabla L_t(\theta)\|^2] \leq \sigma^2.$$

**Assumption 4.4.** *(Bounded Initialization Error) Assume $\forall \theta^* \in \arg\min_{\theta \in \mathbb{R}^d} L(\theta)$, $\exists B$ such that the initialization $\theta_0$ satisfies $\|\theta_0 - \theta^*\| \leq B$.*

To facilitate the analysis, we assume that all task objective functions are optimized with the same number of iterations, denoted $K_t = K \, \forall t \in [T]$, and that they use constant learning rates $\eta_t^k = \eta \, \forall t \in [T]$ and $\forall k \in [K]$. Additionally, we set the outer step size to $\beta = 1$, reducing Task Arithmetic to model averaging.

Although there is no universal definition of data heterogeneity, several notions are commonly referenced in the literature (Li et al., 2019; Karimireddy et al., 2020; Khaled et al., 2020; Koloskova et al., 2020; Woodworth et al., 2020b;a; Glasgow et al., 2022; Patel et al., 2024; Woodworth et al., 2021). One widely adopted first-order notion of data heterogeneity is given by the following assumption (Koloskova et al., 2020; Patel et al., 2024; Woodworth et al., 2020b; Glasgow et al., 2022).

**Assumption 4.5.** *(Bounded First-Order Data Heterogeneity at Optima) A set of objective functions $\{L_t\}_{t=1}^T$ satisfies the bounded first-order heterogeneity at optima if $\forall \theta^* \in \arg\min_{\theta \in \mathbb{R}^d} L(\theta)$, $\exists \zeta_*$ such that*

$$\frac{1}{T} \sum_{t=1}^T \|\nabla L_t(\theta^*)\|^2 \leq \zeta_*^2.$$

The quantity $\zeta_*^2$ measures the diversity among the set of functions $\{L_t\}_{t=1}^T$ at the optima of the averaged multi-task objective function $L(\theta)$. Here, the notion of data heterogeneity is defined through objective functions, while Ortiz-Jimenez et al. (2023) defines the Task Arithmetic property from the perspective of network functions. In Appendix A, we further explore the connection between this notion of data heterogeneity and the Task Arithmetic property proposed in Ortiz-Jimenez et al. (2023).

Using the notation from Patel et al. (2024), we define a learning problem satisfying Assumptions 4.2, 4.3, 4.4, and 4.5 as belonging to class $\mathcal{P}_{\zeta_*}^{H,B,\sigma}$. Based on the upper bound from Koloskova et al. (2020) and the lower bound from Patel et al. (2024), the following theorem characterizes the convergence of one-shot FedAvg.

**Theorem 4.6.** *Assume there is only one communication round $R = 1$. Then for any $K \geq 2, T, H, B, \sigma, \zeta_*^2$,*

$$\min_{\{L_t\}_{t=1}^T \in \mathcal{P}_{\zeta_*}^{H,B,\sigma}} \mathbb{E}[L(\theta_{OS})] - L(\theta^*) \succeq HB^2 + \frac{(H\sigma^2 B^4)^{1/3}}{K^{1/3}} + \frac{\sigma B}{\sqrt{TK}} + (H\zeta_*^2 B^4)^{1/3} \tag{5}$$

*and*

$$\max_{\{L_t\}_{t=1}^T \in \mathcal{P}_{\zeta_*}^{H,B,\sigma}} \mathbb{E}[L(\theta_{OS})] - L(\theta^*) \preceq HB^2 + \frac{(H\sigma^2 B^4)^{1/3}}{K^{1/3}} + \frac{\sigma B}{\sqrt{TK}} + (H\zeta_*^2 B^4)^{1/3}. \tag{6}$$

Here, $\succeq$ and $\preceq$ denote inequalities that hold up to absolute constants. Based on the above theorem, we make several observations about Task Arithmetic. First, data heterogeneity $\zeta_*^2$ degrades the performance of Task Arithmetic. The term $(H\zeta_*^2 B^4)^{1/3}$ is a non-vanishing error term introduced by $\zeta_*^2$, highlighting the negative impact of data heterogeneity.

Second, the one-shot learning nature of Task Arithmetic presents challenges that limit its performance. Notably, another non-vanishing term in Theorem 4.6, $HB^2$, arises due to the one-shot learning setup. In contrast, for FedAvg with $R$ communication rounds, this term becomes $\frac{HB^2}{R}$ and diminishes as the number of communication rounds $R$ increases. Moreover, although both $\frac{(H\sigma^2 B^4)^{1/3}}{K^{1/3}}$ and $\frac{\sigma B}{\sqrt{TK}}$ decrease as the number of local steps $K$ grows, they decay much more slowly in the one-shot setting compared to $R$ rounds of FedAvg. With multiple communication rounds, these terms are given by $\frac{(H\sigma^2 B^4)^{1/3}}{K^{1/3} R^{2/3}}$ and $\frac{\sigma B}{\sqrt{TKR}}$ respectively (Patel et al., 2024). This underscores the additional challenges introduced by the one-shot learning paradigm.

Third, starting with a good pre-trained model is important. The influence of pre-training is captured by the term $B$ introduced in Assumption 4.4. This quantity $B$ is a critical factor as it appears in every error term, particularly in the non-vanishing term $(H\zeta_*^2 B^4)^{1/3}$. Starting with a well-suited pre-trained model that has a smaller $B$ significantly mitigates the adverse effects of high data heterogeneity $\zeta_*^2$, as a smaller $B$ counteracts the heterogeneity. In fact, the significance of pre-trained models has been observed in experiments of both Task Arithmetic (Ortiz-Jimenez et al., 2023) and Federated Learning (Nguyen et al., 2022; Chen et al., 2022).

*Remark* 4.7. The scaling coefficient $\lambda$ is important. As mentioned before, $\lambda = \frac{\beta}{T}$, meaning that the scaling coefficient depends directly on the outer step size. Although in this section we assume $\beta = 1$ which yields $\lambda = \frac{1}{T}$ and reduces Task Arithmetic to model averaging for simplicity, proper tuning of the scaling coefficient $\lambda$ is essential. Research indicates that the choice of $\beta$ has a significant impact on FedAvg performance (Patel et al., 2024; Karimireddy et al., 2020; Charles & Konečný, 2020; Jhunjhunwala et al., 2023; Malinovsky et al., 2023; Li et al., 2023b). A similar sensitivity to $\lambda$ has been observed in Task Arithmetic: the performance of the final model depends heavily on $\lambda$. For instance, Figure 15 in Ilharco et al. (2023) illustrates how Task Arithmetic's performance can vary dramatically with changes to $\lambda$.

## 4.2 Training Heterogeneity

This subsection examines the effect of training heterogeneity on Task Arithmetic, which we define as follows:

**Definition 4.8.** *(Training Heterogeneity) In Task Arithmetic, training heterogeneity refers to the phenomenon where different task objective functions $L_t$ are optimized with varying local learning rates and numbers of iterations during local training.*

In the previous section, we assumed that all task objective functions are optimized with a fixed number of iterations $K$ and constant learning rate $\eta$. However, in practice, each task objective function is often optimized with different hyperparameter settings, which introduces training heterogeneity and can lead to objective inconsistency (Wang et al., 2020). Now, we extend the setting in Section 4.1 to consider each task objective function $L_t$ being optimized with distinct hyperparameters $\eta_t^{(k)}$ and $K_t$. We adopt notation and apply theoretical insights from Wang et al. (2020) to illustrate the impact of training heterogeneity.

First, let $g_t$ be the stochastic gradient of $L_t$, and we define the following matrix of stochastic gradients for each task $t$

$$G_t = [g_t(\theta_t^{(0)}) \quad g_t(\theta_t^{(1)}) \quad \dots \quad g_t(\theta_t^{(K_t-1)})] \in \mathbb{R}^{d \times K_t}.$$

Next we define the vector of normalized learning rates for each task $t$ as

$$a_t = [\frac{\eta_t^{(0)}}{\eta}, \frac{\eta_t^{(1)}}{\eta}, \ldots, \frac{\eta_t^{(K_t-1)}}{\eta}]^T \in \mathbb{R}^{K_t}$$

where $\eta$ is a constant used to normalize the learning rates, whose purpose will be specified later. Using this notation, we can rewrite equation (4) for $\theta_{TA}$ as follows:

$$\theta_{TA} = \theta_0 - \lambda \sum_{t=1}^{T} \eta G_t a_t = \theta_0 - \frac{\beta}{T} \sum_{t=1}^{T} \eta \|a_t\|_1 \frac{G_t a_t}{\|a_t\|_1} \tag{7}$$

where the second equality follows from $\lambda = \frac{\beta}{T}$ as mentioned in Section 3. Next, we denote $\tau_{\text{eff}} = \frac{\beta}{T} \sum_{t=1}^{T} \|a_t\|_1$ as the effective number of steps which measures the average number of updates accumulated using the constant learning rate $\eta$, and denote $w_t = \frac{\|a_t\|_1}{\sum_{s=1}^{T} \|a_s\|_1}$ as the aggregation weight for task $t$. Then we can further rewrite equation (7) as

$$\theta_{TA} = \theta_0 - \tau_{\text{eff}} \sum_{t=1}^{T} \eta w_t \frac{G_t a_t}{\|a_t\|_1}. \tag{8}$$

Notice the weight coefficients vector $[w_1; w_2; \ldots; w_T]$ differs from the original uniform coefficients $[\frac{1}{T}; \frac{1}{T}; \ldots; \frac{1}{T}]$ in the objective function $L$ from equation (1). This discrepancy is caused by training heterogeneity, and leads FedAvg with multiple communication rounds to converge to the stationary point of a different objective function $\tilde{L}(\theta) := \sum_{t=1}^{T} w_t L_t(\theta)$ which is inconsistent with the original objective function $L$. While Task Arithmetic involves only one round of FedAvg, the inconsistency still remains due to training heterogeneity. Formally, we present the following assumptions and adapt Theorems 1 and 2 from Wang et al. (2020) to illustrate this inconsistency in our setting.

**Assumption 4.9.** *(Smoothness) Assume all the task objective functions $L_t$ are $H$-smooth. That is, $\forall t \in [T]$ and $\forall \theta, \varphi \in \mathbb{R}^d$, $\|\nabla L_t(\theta) - \nabla L_t(\varphi)\| \leq H\|\theta - \varphi\|$.*

**Assumption 4.10.** *(Bounded Gradient Heterogeneity) For any set of weights $\{w_t\}_{t=1}^{T}$ such that $\sum_{t=1}^{T} w_t = 1$, there exist constants $\alpha$ and $\zeta$ such that $\forall \theta \in \mathbb{R}^d$,*

$$\sum_{t=1}^{T} w_t \|\nabla L_t(\theta)\|^2 \leq \alpha^2 \|\sum_{t=1}^{T} w_t \nabla L_t(\theta)\|^2 + \zeta^2.$$

*Remark* 4.11. Notice that Assumption 4.10 imposes a more restrictive condition on data heterogeneity compared to Assumption 4.5. Currently, no unified notion of data heterogeneity exists for Federated Learning. Since this section focuses on training heterogeneity, we adopt this more restrictive notion of data heterogeneity, as done in Wang et al. (2020), to facilitate theoretical development.

**Theorem 4.12.** *(Theorem 1 and 2 from (Wang et al., 2020)) Consider $\theta_{TA}$ from update rule (7). Denote $\tilde{L}(\theta) = \sum_{t=1}^{T} w_t L_t(\theta)$ and $\bar{K} = \frac{1}{T} \sum_{t=1}^{T} K_t$. Let $\eta = \sqrt{T/\bar{K}}$. Under Assumption 4.3, 4.9 and 4.10, we have the following bound on the gradient norm $\|\nabla \tilde{L}(\theta_{TA})\|^2$:*

$$\mathbb{E}[\|\nabla \tilde{L}(\theta_{TA})\|^2] \leq \frac{4(\tilde{L}(\theta_0) - \tilde{L}_{\inf})(\bar{K}/\tau_{eff})}{\sqrt{T\bar{K}}} + \frac{4H\sigma^2 A_1}{\sqrt{T\bar{K}}} + \frac{6TH^2\sigma^2 A_2}{\bar{K}} + \frac{12TH^2\zeta^2 A_3}{\bar{K}}. \tag{9}$$

*Specifically, $\tilde{L}_{\inf} = \inf_\theta \tilde{L}(\theta)$, $A_1 = \tau_{eff}T \sum_{t=1}^{T} \frac{w_t^2\|a_t\|_2^2}{\|a_t\|_1^2}$, $A_2 = \sum_{t=1}^{T} w_t(\|a_t\|_2^2 - a_{t,-1}^2)$ and $A_3 = \max_t\{\|a_t\|_1(\|a_t\|_1 - a_{t,-1})\}$ where $a_{t,-1}$ denotes the last coordinate of the vector $a_t$. Denote the RHS of inequality (9) as $\epsilon$. Moreover, we have the following bound on gradient norm $\|\nabla L(\theta_{TA})\|^2$:*

$$\mathbb{E}[\|\nabla L(\theta_{TA})\|^2] \leq 2[\chi_{p||w}^2(\alpha^2 - 1) + 1]\epsilon + 2\chi_{p||w}^2\zeta^2 \tag{10}$$

*where $\chi_{p||w}^2 = \sum_{t=1}^{T} \frac{(\frac{1}{T} - w_t)^2}{w_t}$ is the chi-square divergence between the weight coefficient vectors $p = [\frac{1}{T} \ \frac{1}{T} \ \ldots \ \frac{1}{T}]$ and $w = [w_1 \ w_2 \ \ldots \ w_T]$.*

The theorem above illustrates how heterogeneous local training affects the convergence of Task Arithmetic to a stationary point of $L$. The convergence rate in (10) is influenced by two key factors: the term $\chi^2_{p||w}\zeta^2$ and the convergence rate to a stationary point of $\tilde{L}$ in (9). For $\chi^2_{p||w}\zeta^2$, when different training processes are used for each objective function, $\chi^2_{p||w}$ is non-zero, resulting in $\chi^2_{p||w}\zeta^2$ as a persistent error term. With training heterogeneity, $\chi^2_{p||w}\zeta^2$ only vanishes if $\zeta^2 = 0$, indicating minimal data heterogeneity. This highlights the interaction between data heterogeneity and training heterogeneity: significant data heterogeneity exacerbates the negative effects of training heterogeneity, intensifying the overall performance degradation.

When all task objective functions are optimized with the same number of iterations $K$ and a consistent learning rate schedule $\{\eta^{(0)}, \ldots, \eta^{(K-1)}\}$, we have $w_t = \frac{1}{T}$. This yields $\chi^2_{p||w} = 0$ and $L = \tilde{L}$, aligning the actual objective function $\tilde{L}$ being optimized with the original objective function $L$. In this scenario, objective inconsistency is effectively eliminated. The convergence rate in (10) is reduced to (9).

*Remark* 4.13. In Theorem 4.12, unlike in Theorem 4.6, we make no assumptions about the convexity of the objective functions, which naturally results in a looser convergence rate. Since the primary focus of this paper is not on deriving a tighter convergence bound for non-convex settings, we limit our analysis to applying existing theoretical results to understand the behavior of Task Arithmetic.

## 5 Adapting Federated Learning Algorithms for Task Arithmetic

In the previous section, we used insights from FedAvg to analyze how data and training heterogeneity negatively impact Task Arithmetic. In order to address these challenges, numerous algorithms have been developed to improve FedAvg for more efficient Federated Learning (see Section 2 for related work). Thus, we adapt some Federated Learning algorithms to solve heterogeneity challenges in Task Arithmetic for better model merging performance. Selecting the right Federated Learning algorithms to implement requires a clear understanding of key challenges that complicate the adaptation. In this section, we begin with analyzing several key challenges.

### 5.1 Challenges in Adapting Federated Learning Algorithms for Task Arithmetic

First, the number of communication rounds is limited. Since Task Arithmetic is only one-shot Federated Learning, algorithms relying on multiple communication rounds are unsuitable. For instance, some Federated Learning algorithms add regularization terms to local objective functions (Li et al., 2020b; Durmus et al., 2021) to encourage local updates to remain close to the global model parameters transmitted from the previous communication round. However, in our one-shot setting, only the pre-trained model parameters $\theta_0$ are communicated. Applying such regularization would force each task's fine-tuned parameters to remain close to $\theta_0$, potentially hindering both convergence and task-specific performance. Similarly, algorithms that address heterogeneity through iterative techniques such as variance reduction (Karimireddy et al., 2020; Reddi et al., 2020) or adaptive step size tuning (Jhunjhunwala et al., 2023; Li et al., 2023b) require the accumulation and updating of certain metrics over multiple communication rounds. Since Task Arithmetic operates within a one-shot setting, implementing such iterative updates is impossible.

Second, no additional training is allowed. To counteract the effects of heterogeneity or limited communication and data budget, some Federated Learning algorithms require additional training, imposing additional computational cost. For instance, Ye et al. (2023b) ask each device to learn metrics that compare local and global data through training, and these metrics are then used as additional scores for aggregation. Zhang et al. (2022) train an additional generator to generate data which is subsequently used to train a global model in one-shot Federated Learning setting.

Third, no additional datasets are available. Many Federated Learning algorithms rely on supplementary datasets, which is not feasible for modifying Task Arithmetic. In one-shot Federated Learning, a common approach to address data heterogeneity is knowledge distillation (Zhou et al., 2020; Guha et al., 2019; Li et al., 2020a). These methods often require access to extra datasets from which either local devices or the central server distills knowledge to improve model performance.

Given the constraints and unique needs for adapting Federated Learning algorithms, we identify the following three criteria for selecting Federated Learning algorithms: adaptability to one-shot setting, no additional training required and no access to additional datasets required.

## 5.2 Algorithms

With criteria established above, we now explore four Federated Learning algorithms FedNova (Wang et al., 2020), FedGMA (Tenison et al., 2022), Median (Yin et al., 2018) and CCLIP (Karimireddy et al., 2021) that align with these guidelines, explaining their motivations and how they modify Task Arithmetic. For more detailed explanation and further understanding of these algorithms, please refer to the original papers.

**FedNova (Wang et al., 2020)**  FedNova addresses objective inconsistency caused by training heterogeneity by replacing the heterogeneous weight vector $[w_1 \ w_2 \ \ldots \ w_T]$ with the uniform weight vector $[\frac{1}{T} \ \frac{1}{T} \ \ldots \ \frac{1}{T}]$, ensuring consistent weighting across tasks. This approach adapts easily to the one-shot setting. Using the notation from Section 4.2, FedNova modifies the Task Arithmetic update as:

$$\theta_{TA} = \theta_0 - \tau_{\text{eff}} \sum_{t=1}^{T} \frac{\eta}{T} \frac{G_t a_t}{\|a_t\|_1} = \theta_0 + \lambda(\frac{1}{T} \sum_{t=1}^{T} \|a_t\|_1) \sum_{t=1}^{T} \frac{\tau_t}{\|a_t\|_1} \tag{11}$$

where $\tau_{\text{eff}} = \frac{\beta}{T} \sum_{t=1}^{T} \|a_t\|_1$ is the effective number of steps defined in Section 4.2, $\tau_t = -\eta G_t a_t$ is the task vector, and $\lambda = \frac{\beta}{T}$ is the scaling coefficient. In other words, FedNova normalizes each task vector by $\|a_t\|_1$ and rescales the scaling coefficient by the average $\frac{1}{T} \sum_{t=1}^{T} \|a_t\|_1$.

**FedGMA (Tenison et al., 2022)**  FedGMA addresses data heterogeneity by mitigating sign conflicts among local updates. Such sign conflicts can cause information loss and slower convergence. FedGMA uses a gradient mask to reduce the impact of conflicting directions and preserves meaningful information. Specifically, it computes an agreement score $A$ to measure alignment across task vectors $\{\tau_t\}_{t=1}^{T}$. Based on a threshold $\rho$, FedGMA constructs a mask $\tilde{M}$ that emphasizes coordinates with strong agreement while reducing the influence of others. Formally, define $A = \left| \frac{1}{T} \sum_{t=1}^{T} \text{sign}(\tau_t) \right|$ and $\tilde{M}_j = \begin{cases} 1, & \text{if } A_j \geq \rho \\ A_j, & \text{otherwise} \end{cases}$ where $j$ denotes the $j$-th coordinate and $\text{sign}(\cdot)$ is applied in a coordinate-wise manner. This yields

$$\theta_{TA} = \theta_0 + \lambda \tilde{M} \odot \sum_{t=1}^{T} \tau_t. \tag{12}$$

**Median (Yin et al., 2018)**  Coordinate-Wise Median (Yin et al., 2018), originally designed to handle adversarial updates in Federated Learning, is adapted here to address data and training heterogeneity in Task Arithmetic. Due to diverse data distributions or differing hyperparameter settings, some task vectors may have extreme values. By selecting the median value for each coordinate, this method reduces the influence of outliers while maintaining overall performance across tasks. It modifies Task Arithmetic as

$$\theta_{TA} = \theta_0 + \lambda \, \text{med}(\tau_1, \ldots, \tau_T) \tag{13}$$

where $\text{med}(\cdot)$ computes the coordinate-wise median of $\{\tau_t\}_{t=1}^{T}$.

**CCLIP (Karimireddy et al., 2021)**  CCLIP, short for centered clipping, is another widely applied robust aggregation method towards adversarial devices in Federated Learning. With the same motivation to using the Coordinate-Wise Median, we use CCLIP to reduce the impact of extreme task vectors. CCLIP is implemented with a predefined threshold $\rho$ and modifies Task Arithmetic as follows:

$$\theta_{TA} = \theta_0 + \lambda \sum_{t=1}^{T} \tau_t \min\{1, \frac{\rho}{\|\tau_t\|}\}. \tag{14}$$

When the norm of a task vector $\|\tau_t\|$ exceeds the threshold $\rho$, this method identifies it as an outlier and shrinks its magnitude by a factor of $\frac{\rho}{\|\tau_t\|}$.

# 6 Experiments on Merging CLIP

In this section, we present and discuss our experimental results on CLIP-ViT-B-32 (Radford et al., 2021) for image classifications. We follow the same experimental paradigm as (Ilharco et al., 2023). Specifically, we use CLIP-ViT-B-32 (Radford et al., 2021) as the pre-trained model and eight datasets: Cars (Krause et al., 2013), DTD (Cimpoi et al., 2014), EuroSAT (Helber et al., 2019), GTSRB (Stallkamp et al., 2011), MNIST (LeCun, 1998), RESISC45 (Cheng et al., 2017), SUN397 (Xiao et al., 2016), and SVHN (Netzer et al., 2011), to construct eight task vectors.

In total, there are 247 ways to select $T$ different task vectors from these eight task vectors where $T \in [2, 8]$: $247 = \sum_{T=2}^{8} \binom{8}{T}$. For each algorithm, we therefore conduct 247 experiments. In each experiment, we merge $T$ selected task vectors and evaluate on the $T$ datasets corresponding to the task vectors used. Our evaluation metric is normalized accuracy (Ilharco et al., 2023), defined as the test accuracy normalized by the fine-tuned model's accuracy. That is,

$$\text{normalized accuracy on task } t = \frac{\text{accuracy on task } t}{\text{accuracy of the fine-tuned model } t \text{ on task } t}.$$

## 6.1 Experimental Results

In the first part of the experiments (Section 6.1.1), to simulate practical conditions of training heterogeneity, we fine-tune CLIP-ViT-B-32 on each dataset using three learning rates $\{1e{-}4, 1e{-}5, 1e{-}6\}$ and different numbers of iterations. Then we select the best fine-tuned checkpoints via cross-validation on validation datasets. Refer to Appendix B.1 for further details on fine-tuning and cross-validation.

In the second part of the experiments (Section 6.1.2), to better understand the impact of training heterogeneity, we use the task vectors provided by Ilharco et al. (2023), which were fine-tuned with the same number of iterations and the same learning rates, thereby eliminating training heterogeneity.

### 6.1.1 Merging with training heterogeneity

We first report experimental results using task vectors fine-tuned with training heterogeneity. Table 1 summarizes the performance of various methods in a specific experimental setup: merging all eight task vectors, corresponding to the scenario where $T = 8$. We report the average normalized accuracy as well as the normalized accuracy for each dataset. Task Arithmetic is used as the baseline method for comparison. As shown in the table, all four adapted Federated Learning methods outperform the baseline by a substantial margin. Moreover, we observe that Median and CCLIP yield the most improvement.

| Methods | Average Normalized Accuracy | DTD | EuroSAT | GTSRB | SUN397 | SVHN | MNIST | Cars | RESISC45 |
|---|---|---|---|---|---|---|---|---|---|
| Task Arithmetic | 67.33 | 57.43 | 53.86 | 41.00 | 82.40 | **78.58** | 87.76 | 71.94 | 65.72 |
| Median | 74.55 (↑ 7.22) | **67.51** | **78.05** | 67.12 | 84.02 | 56.69 | 91.32 | **77.51** | **74.17** |
| FedNova | 69.57 (↑ 2.24) | 57.08 | 50.37 | 61.47 | **86.62** | 77.68 | 85.18 | 74.22 | 63.96 |
| FedGMA | 68.55 (↑ 1.22) | 60.01 | 58.69 | 45.02 | 84.13 | 71.19 | 86.65 | 74.53 | 68.13 |
| CCLIP | **74.82** (↑ 7.49) | 66.76 | 75.42 | **73.87** | 83.18 | 58.51 | **92.03** | 76.40 | 72.39 |

Table 1: **Merging all eight task vectors using five different methods.** Each method is evaluated on eight datasets, with normalized accuracy reported for each dataset. The highest and second-highest normalized accuracy values for each dataset are highlighted in bold and underlined, respectively.

Table 2 and Figure 1 summarize the performance comparison between Task Arithmetic and other Federated Learning algorithms across 247 experiments. In Table 2, we report the percentage of 247 experiments in which the average normalized accuracy improves, remains unchanged, or degrades when using four Federated Learning methods compared to the baseline, Task Arithmetic. The average normalized accuracy is calculated by averaging over the number of task vectors being used. In order to better visualize the performance differences for each method, in Figure 1, we use histograms to show the frequencies of experiments within

| Methods | Percentage of Improved Experiments Compared to Task Arithmetic | Percentage of Unchanged Experiments Compared to Task Arithmetic | Percentage of Degraded Experiments Compared to Task Arithmetic |
|---|---|---|---|
| Median | 67.61% | 0% | 32.39% |
| FedNova | 63.56% | 0% | 36.44% |
| FedGMA | 40.49% | 18.22% | 41.29% |
| CCLIP | 91.50% | 0% | 8.5% |

Table 2: **Percentage of improved, unchanged, and degraded experiments using different methods compared to Task Arithmetic.**

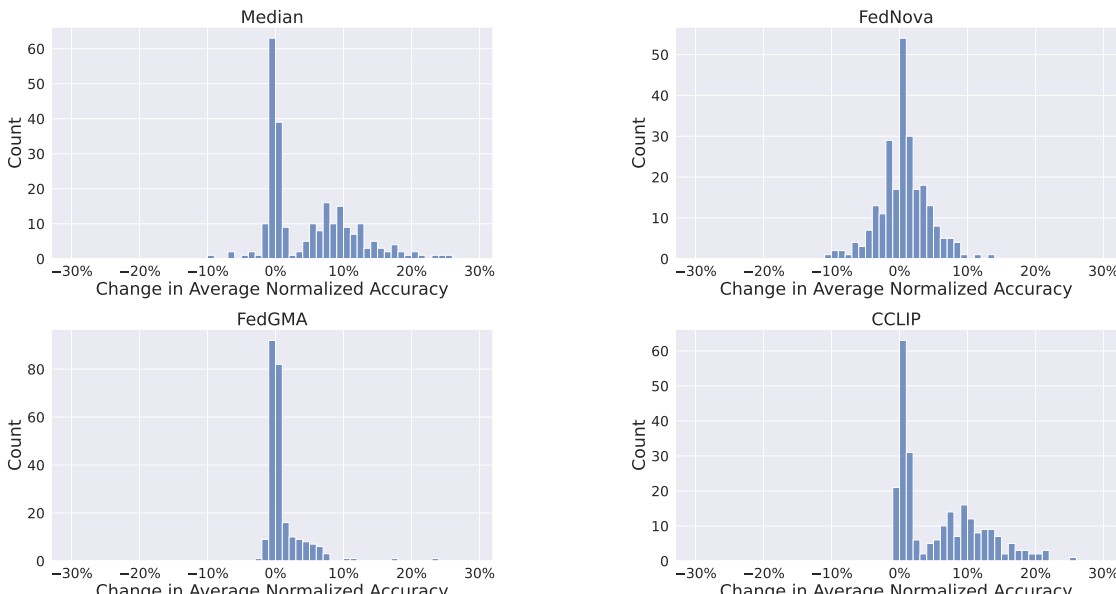

Figure 1: **Histograms showing the change in average normalized accuracy for four different methods compared to Task Arithmetic.** The x-axis shows the difference in average normalized accuracy relative to Task Arithmetic, with positive values indicating improvement and negative values indicating degradation. The y-axis represents the number of experiments within each range of change values.

each range of change in average normalized accuracy. Median, FedNova, and CCLIP consistently show the ability to improve upon Task Arithmetic in most cases, while FedGMA typically demonstrates either no change or slight improvements.

### 6.1.2 Merging without training heterogeneity

In Section 4.2, we analyzed how training heterogeneity causes objective inconsistency, degrading Task Arithmetic's performance. While in the experiments conducted by Ilharco et al. (2023), all task vectors were homogeneously fine-tuned using a consistent learning rate 1e−5, our approach in Section 6.1.1 employs heterogeneous fine-tuning. We compare the performance of Task Arithmetic on these two sets of task vectors to validate our theoretical findings in Section 4.2.

Table 3 compares Task Arithmetic's performance on heterogeneously and homogeneously fine-tuned task vectors when merging all eight task vectors. Again we report the average normalized accuracy and the normalized accuracy for each dataset. As evident from the table, Task Arithmetic with homogeneous fine-tuning consistently outperforms its heterogeneous counterpart across all datasets, except for SUN397.

Table 4 and Figure 2 compare Task Arithmetic's performance on homogeneously and heterogeneously fine-tuned task vectors across 247 experiments. Homogeneous fine-tuning outperforms heterogeneous fine-tuning in 92.31% of cases, as shown in Table 4. Moreover, Figure 2 shows that homogeneous fine-tuning can improve

the average normalized accuracy by up to more than 30%. These results highlight the significant negative impact of training heterogeneity on the performance of Task Arithmetic.

| Methods | Average Normalized Accuracy | DTD | EuroSAT | GTSRB | SUN397 | SVHN | MNIST | Cars | RESISC45 |
|---|---|---|---|---|---|---|---|---|---|
| Task Arithmetic with Heterogeneous Fine-Tuning | 67.33 | 57.43 | 53.86 | 41.00 | 82.40 | 78.58 | 87.76 | 71.94 | 65.72 |
| Task Arithmetic with Homogeneous Fine-Tuning | 77.34 (↑ 10.01) | 64.90 | 77.93 | 69.47 | 80.64 | 80.26 | 96.42 | 75.98 | 73.01 |

Table 3: **Using Task Arithmetic to merge eight task vectors from heterogeneous and homogeneous fine-tuning processes.** Each method is evaluated on eight datasets, with normalized accuracy reported for each dataset.

| | Percentage of Improved Experiments | Percentage of Unchanged Experiments | Percentage of Degraded Experiments |
|---|---|---|---|
| Task Arithmetic with Homogeneous Fine-Tuning | 92.31% | 0% | 7.69% |

Table 4: **Percentage of improved, unchanged, and degraded experiments using task vectors with homogeneous fine-tuning process compared to those with heterogeneous fine-tuning process.**

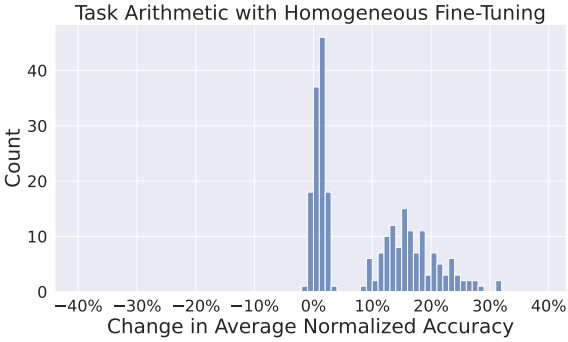

Figure 2: **Histogram showing the change in average normalized accuracy when using task vectors from homogeneous fine-tuning compared to heterogeneous fine-tuning.** The x-axis shows the difference in average normalized accuracy relative to Task Arithmetic, with positive values indicating improvement and negative values indicating degradation. The y-axis represents the number of experiments within each range of change values.

## 6.2 Discussion on Experimental Results

We now discuss a key observation from our experimental results: in practice, training heterogeneity poses a greater challenge than data heterogeneity for Task Arithmetic.

While Section 4.1 highlights how data heterogeneity degrades Task Arithmetic, our experimental results in Section 6.1 show it is less problematic compared to training heterogeneity. First, FedNova, designed to address training heterogeneity, consistently outperforms Task Arithmetic more frequently and significantly than FedGMA which targets data heterogeneity. Table 2 and Figure 1 demonstrate that FedNova both improves performance more frequently and achieves greater overall gains than FedGMA.

Second, among Federated Learning algorithms, CCLIP and Median demonstrate the best performance. As discussed in Section 5, these methods are designed for robust aggregation in the presence of outliers. In our setting, they effectively address training heterogeneity which causes certain task vectors to have

disproportionately large norms and behave like outliers. For example, the cross-validation process selects a much larger learning rate of $1e-4$ for SVHN, compared to $1e-5$ used for other datasets. This hyperparameter setup results in the SVHN task vector having a significantly larger norm (reported in Appendix B.1.5), making it an outlier that negatively impacts the merged model's performance on other tasks when using Task Arithmetic. By employing robust aggregation methods like Median and CCLIP, we reduce the influence of the SVHN task vector, which improves the merged model's performance on other tasks.

Third, when comparing Table 2 and Table 4, we see that homogeneous fine-tuning leads to more frequent improvements over Task Arithmetic compared to the other four algorithms Median, FedNova, FedGMA and CCLIP. Similarly, Figure 2 demonstrates that homogeneous fine-tuning results in the most frequent and substantial positive changes in average normalized accuracy.

Further evidence is presented in Appendix B.2, where we evaluate the performance of Median, FedGMA and CCLIP on task vectors generated via homogeneous fine-tuning. Using the performance of Task Arithmetic on these homogeneously fine-tuned task vectors as the baseline, we find that Federated Learning algorithms rarely improve upon the baseline. In fact, Task Arithmetic consistently emerges as the best-performing approach when merging those task vectors generated without training heterogeneity. This reinforces our observation that training heterogeneity is a more significant issue than data heterogeneity in practice.

## 7 Experiments on Merging LLMs

We now present and discuss our experimental results on merging LLMs for three tasks: instruction following, mathematical reasoning, and code generation. We follow the experimental paradigm of Yu et al. (2024). We merge task vectors from three models—WizardLM-13B (Xu et al., 2023), WizardMath-13B (Luo et al., 2023), and Llama-2-13B-Code-Alpaca (Chaudhary, 2023)—for instruction following, mathematical reasoning, and code generation, respectively. All three models are fine-tuned from Llama2-13B (Touvron et al., 2023). For instruction following, we evaluate the models on AlpacaEval (Li et al., 2023a). For mathematical reasoning, we use GSM8K (Cobbe et al., 2021) and MATH (Hendrycks et al., 2021). For code generation, we use HumanEval (Chen et al., 2021) and MBPP (Austin et al., 2021). Performance metrics include win rate for AlpacaEval, zero-shot accuracy for GSM8K and MATH, and pass@1 for HumanEval and MBPP.

Since all models used in this experiment are downloaded from HuggingFace, we do not have access to their fine-tuning hyperparameter settings. As a result, FedNova cannot be applied in this experiment because it requires knowledge of learning rates and the number of iterations, which are unavailable. Furthermore, when implementing Median, taking the median of two vectors reduces to averaging, which is equivalent to Task Arithmetic. Thus, we implement Median only for merging three task vectors, with the corresponding results deferred to Appendix C.2. For additional details on experiments, please refer to Appendix C.

### 7.1 Experimental Results

In Table 5, we compare the performance of three methods: Task Arithmetic, FedGMA, and CCLIP. The results show that when merging two out of three task vectors, FedGMA and CCLIP often outperform Task Arithmetic. However, when merging all three task vectors, Task Arithmetic shows superior performance on code generation and instruction-following tasks. Notably, Task Arithmetic consistently excels in instruction-following tasks, achieving either the highest accuracy or accuracy comparable to the other methods.

### 7.2 Discussion on Experimental Results

In this section, we discuss a key observation from our experimental results: training heterogeneity arises not only from differences in hyperparameters but also from variations in tuning methods.

In Section 4.2, we theoretically analyzed how using different learning rates and number of iterations creates training heterogeneity and thus leads to objective inconsistency. However, our experimental results in Section 7 reveal that employing different fine-tuning methods further exacerbates training heterogeneity. Specifically, in our experiments, the Llama-2-13B-Code-Alpaca model is fine-tuned using QLoRA (Dettmers et al., 2024), a parameter-efficient fine-tuning (PEFT) approach.

| Tasks | Methods | Mathematical Reasoning | | Code Generation | | Instruction Following |
|---|---|---|---|---|---|---|
| | | GSM8K | MATH | HumanEval | MBPP | AlpacaEval |
| Math Code | Task Arithmetic | 64.22 | **14.1** | 1.22 | 8.66 | / |
| | FedGMA | 65.5 | 12.66 | **15.85** | **21.8** | / |
| | CCLIP | **65.81** | 13.48 | 4.27 | 7.6 | / |
| Instruction Math | Task Arithmetic | 65.88 | 13.32 | / | / | 69.96 |
| | FedGMA | **66.72** | **14.48** | / | / | 62.04 |
| | CCLIP | 64.75 | 13.18 | / | / | **69.99** |
| Instruction Code | Task Arithmetic | / | / | **32.32** | 32.2 | **79.76** |
| | FedGMA | / | / | 20.12 | 26 | 49.55 |
| | CCLIP | / | / | **32.32** | **34.2** | 76.02 |
| Instruction Math Code | Task Arithmetic | 58.45 | 12.06 | **25.16** | **31** | **70.89** |
| | FedGMA | 57.16 | 11.96 | 20.12 | 27.4 | 64.13 |
| | CCLIP | **62.93** | **12.96** | 20.12 | 27.6 | 66.91 |

Table 5: **Performance of merging LLMs.** The best performance for each dataset is highlighted in bold.

PEFT adjusts only a small subset of parameters while leaving the rest unchanged (Han et al., 2024). Consequently, task vectors generated by PEFT typically have smaller norms compared to those generated through standard fine-tuning. In our experiments, Llama-2-13B-Code-Alpaca, which is fine-tuned for code generation using PEFT, produces a task vector with a notably small norm of 5.05. In contrast, WizardLM-13B and WizardMath-13B, fine-tuned for instruction following and mathematical reasoning via standard fine-tuning, generate task vectors with much larger norms of 142.61 and 52.62, respectively. This discrepancy can pose challenges when merging task vectors. Simply regulating the behavior of task vectors with larger norms can lead to unintended negative effects. As shown in Table 5, when merging tasks include both instruction following (large norm) and code generation (small norm), FedGMA and CCLIP either fail to outperform Task Arithmetic or achieve comparable performance on these two tasks. This highlights that addressing training heterogeneity by focusing solely on differences in hyperparameters is insufficient in practice.

While some studies have explored the challenges of merging large models fine-tuned via PEFT (Zhang et al., 2023; Wu et al., 2024), merging PEFT-generated task vectors with those produced by standard fine-tuning remains an open research question. Further investigation is required to design effective strategies for combining such task vectors in Task Arithmetic. Additionally, more research is needed to develop robust aggregation methods in Federated Learning to address this type of practical training heterogeneity.

## 8   Limitations and Conclusion

**Limitations**   First, in Federated Learning, there remains a gap in theoretical understanding and algorithmic development for addressing data heterogeneity in large-scale experiments. This might arise from the lack of a universal notion of data heterogeneity. While existing theoretical work primarily focuses on first-order heterogeneity such as Assumption 4.5, recent studies (Patel et al., 2024; Wang et al., 2022) suggest that this perspective may be insufficient to fully capture data heterogeneity in practice. This underscores the need for more refined definitions of data heterogeneity and the development of corresponding algorithms.

Second, a recent study (Yadav et al., 2024b) shows that when merging very large models such as PaLM-2-64B (Anil et al., 2023), the differences in performance between various merging methods tend to diminish. While Section 4.1 discusses how a high-quality pre-trained model mitigates the adverse effects of data heterogeneity, our framework does not yet provide a theoretical explanation for why the large size of the pre-trained model can also alleviate data heterogeneity, and we leave this as an open question for future work.

**Conclusion**   In this paper, we established a connection between Task Arithmetic and one-shot Federated Learning. By leveraging theoretical insights from Federated Learning, we identified and analyzed two key sources of heterogeneity—data heterogeneity and training heterogeneity—and their impact on Task Arithmetic. Also, we adapted Federated Learning algorithms for model merging, demonstrating their great

potential to significantly improve over the performance of Task Arithmetic. We hope this work serves as a foundation for advancing the understanding, enhancing the algorithms, and expanding the applications of Task Arithmetic through the lens of Federated Learning.

**Author Contributions**

X. Boix ideated the research; Z. Tao conceptualized the theoretical part of the research with contributions from I. Mason and X. Boix; Z. Tao and I. Mason conceptualized the experimental part of the research with contributions from X. Boix; Z. Tao wrote the code and ran the experiments with contributions from I. Mason; Z. Tao analyzed the experimental results with contributions from I. Mason, S. Kulkarni and X. Boix; Z. Tao wrote the paper with contributions from I. Mason, S. Kulkarni and X. Boix; S.Kulkarni and X. Boix supervised the project.

**Acknowledgments**

We thank the reviewers and action editor for their insightful comments and constructive feedback, which have significantly improved the quality of this work. We are also grateful to Kasper Vinken and Mehdi Bahrami for valuable discussions and feedback throughout the project.

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

# A  Task Arithmetic Property

In this section, we review a paper that provides theoretical insights into Task Arithmetic (Ortiz-Jimenez et al., 2023). We examine the relationship between data heterogeneity and the Task Arithmetic property proposed in their work. To facilitate a stronger connection between their framework and our perspective, we adapt their definition of the Task Arithmetic property as follows.

**Property A.1.** *(Task Arithmetic Property 1 from Ortiz-Jimenez et al. (2023)) Consider a set of task vectors $\{\tau_t\}_{t=1}^T$ with associated task data distributions $\{\mathcal{D}_t\}_{t=1}^T$. Suppose all the distributions $\{\mathcal{D}_t\}_{t=1}^T$ have non-intersecting supports. Let $f$ be a network function. We say a network function $f$ satisfies the Task Arithmetic property around $\theta_0$ with respect to $\{\tau_t\}_{t=1}^T$ and $\{\mathcal{D}_t\}_{t=1}^T$ if*

$$f(x, \theta_0 + \lambda \sum_{t=1}^T \tau_t) = f(x, \theta_0 + \tau_t) \forall x \in \operatorname{supp}(\mathcal{D}_t)$$

*and*

$$f(x, \theta_0 + \lambda \sum_{t=1}^T \tau_t) = f(x, \theta_0) \forall x \notin \cup_{t=1}^T \operatorname{supp}(\mathcal{D}_t).$$

Notice that the Task Arithmetic property is defined through the network function $f$, while Assumption 4.5 for data heterogeneity is defined through the objective functions $L_t$. Recall that $L_t(\theta) = \mathbb{E}_{(x_t,y_t) \sim \mathcal{D}_t}[\ell(\theta; x_t, y_t)]$. The objective function is related to the network function $f$ as follows:

$$L_t(\theta) = \mathbb{E}_{(x_t,y_t) \sim \mathcal{D}_t}[\ell(f(x_t, \theta), y_t)]. \tag{15}$$

This connection highlights how the objective function depends on the underlying network function $f$. Notice that the Task Arithmetic property is a property of the network function $f$, but it does not guarantee the effectiveness of Task Arithmetic. For example, if $\theta_0 + \tau_t$ is far from being optimal for objective function $L_t$, then the performance of $f(x, \theta_0 + \lambda \sum_{t=1}^T \tau_t)$ will also be significantly suboptimal. In the subsequent analysis, we will demonstrate that data heterogeneity serves as a necessary condition for Task Arithmetic achieving optimal performance, assuming $\theta_0 + \tau_t$ is optimal for $L_t$.

**Proposition A.2.** *(Necessary Condition) Suppose the Task Arithmetic property holds true and all the objective functions $L_t$ are $H$-smooth and convex in $\theta$. Further assume $\theta_0 + \tau_t$ is optimal for $L_t$, i.e., $\nabla_\theta L_t(\theta_0 + \tau_t) = 0$. Then $\theta_0 + \lambda \sum_{t=1}^T \tau_t$ is optimal for $L$, and the data heterogeneity at $\theta_0 + \lambda \sum_{t=1}^T \tau_t$ is zero, i.e.,*

$$\frac{1}{T} \sum_{t=1}^T \|\nabla L_t(\theta_0 + \lambda \sum_{t=1}^T \tau_t)\|^2 = 0.$$

*Proof.*

$$\|\nabla L(\theta_0 + \lambda \sum_{t=1}^T \tau_t)\| \leq \frac{1}{T} \sum_{t=1}^T \|\nabla L_t(\theta_0 + \lambda \sum_{t=1}^T \tau_t)\|$$

$$= \frac{1}{T} \sum_{t=1}^T \|\nabla \mathbb{E}_{(x_t,y_t) \sim \mathcal{D}_t}[\ell(f(x_t, \theta_0 + \lambda \sum_{t=1}^T \tau_t), y_t)]\| \quad \text{by equation (15)}$$

$$= \frac{1}{T} \sum_{t=1}^T \|\nabla \mathbb{E}_{(x_t,y_t) \sim \mathcal{D}_t}[\ell(f(x_t, \theta_0 + \tau_t), y_t)]\| \quad \text{by Task Arithmetic property}$$

$$= \frac{1}{T} \sum_{t=1}^T \|\nabla L_t(\theta_0 + \tau_t)\|$$

$$= 0 \quad \text{by the optimality of } \theta_0 + \tau_t.$$

Therefore, $\theta_0 + \lambda \sum_{t=1}^{T} \tau_t$ is also optimal for $L$. Next,

$$
\frac{1}{T} \sum_{t=1}^{T} \|\nabla L_t(\theta_0 + \lambda \sum_{t=1}^{T} \tau_t)\|^2
$$

$$
= \frac{1}{T} \sum_{t=1}^{T} \|\nabla L_t(\theta_0 + \lambda \sum_{t=1}^{T} \tau_t) - \nabla L_t(\theta_0 + \tau_t)\|^2 \quad \text{since } \nabla L_t(\theta_0 + \tau_t) = 0
$$

$$
\overset{(i)}{\leq} \frac{2H}{T} \sum_{t=1}^{T} L_t(\theta_0 + \lambda \sum_{t=1}^{T} \tau_t) - L_t(\theta_0 + \tau_t) + \langle \nabla L_t(\theta_0 + \tau_t), \tau_t - \lambda \sum_{t=1}^{T} \tau_t \rangle
$$

$$
= \frac{2H}{T} \sum_{t=1}^{T} \mathbb{E}_{(x_t,y_t) \sim \mathcal{D}_t}[\ell(f(x_t, \theta_0 + \lambda \sum_{t=1}^{T} \tau_t), y_t)] - \mathbb{E}_{(x_t,y_t) \sim \mathcal{D}_t}[\ell(f(x_t, \theta_0 + \tau_t), y_t)]
$$

$$
\overset{(ii)}{=} \frac{2H}{T} \sum_{t=1}^{T} \mathbb{E}_{(x_t,y_t) \sim \mathcal{D}_t}[\ell(f(x_t, \theta_0 + \tau_t), y_t)] - \mathbb{E}_{(x_t,y_t) \sim \mathcal{D}_t}[\ell(f(x_t, \theta_0 + \tau_t), y_t)]
$$

$$
= 0.
$$

Note that inequality (i) follows from the property that for any $H$-smooth and convex function $L$, $\frac{1}{2H}\|\nabla L(x) - \nabla L(y)\|^2 \leq L(y) - L(x) + \langle \nabla L(x), x - y \rangle$, and equality (ii) follows from Task Arithmetic property. Therefore, the data heterogeneity at $\theta_0 + \lambda \sum_{t=1}^{T}$ is zero. $\square$

The above proposition shows that data heterogeneity being zero is actually a necessary condition for the optimal performance of Task Arithmetic. In other words, the parameters generated by Task Arithmetic have to be a shared optimum for all $L_t$.

## B Merging CLIP

### B.1 Additional Experiment Details on Merging CLIP

In this section, we provide details on the hyperparameter searching process for experiments using CLIP. All experiments on CLIP were conducted on a single NVIDIA V100 GPU.

#### B.1.1 Fine-tuning

For each dataset, we fine-tuned ViT-B-32 using three different learning rates combined with four different numbers of epochs, resulting in a total of 12 distinct hyperparameter configurations per dataset. The selected numbers of epochs were chosen to roughly correspond to training for 1000, 2000, 3000 and 4000 iterations, assuming a batch size of 128 for each dataset. To determine the optimal hyperparameter configuration, we used the validation accuracy to select the best combination of learning rate and epochs. Table 6 summarizes the fine-tuning hyperparameters and cross-validation details.

| Datasets | Learning Rates | Epochs | Best Hyperparameters Configuration |
|----------|----------------|--------|-------------------------------------|
| DTD | {1e-4, 1e-5, 1e-6} | {38, 76, 114, 152} | {1e-5, 114} |
| GTSRB | {1e-4, 1e-5, 1e-6} | {6, 11, 17, 22} | {1e-5, 6} |
| SUN397 | {1e-4, 1e-5, 1e-6} | {7, 14, 21, 28} | {1e-5, 7} |
| MNIST | {1e-4, 1e-5, 1e-6} | {3, 5, 8, 10} | {1e-5, 8} |
| SVHN | {1e-4, 1e-5, 1e-6} | {2, 4, 6, 8} | {1e-4, 6} |
| EuroSAT | {1e-4, 1e-5, 1e-6} | {6, 12, 18, 24} | {1e-5, 18} |
| Cars | {1e-4, 1e-5, 1e-6} | {18, 35, 53, 70} | {1e-5, 35} |
| RESISC45 | {1e-4, 1e-5, 1e-6} | {8, 15, 23, 30} | {1e-5, 23} |

Table 6: **Fine-tuning and cross-validation details for CLIP ViT-B-32**

### B.1.2 Scaling coefficient

To determine the optimal scaling coefficient $\lambda$, we searched the range $[0.05, 0.1, 0.15, \ldots, 1.95, 2.0]$, selecting the value that produces the highest average normalized accuracy in the validation data sets.

### B.1.3 Hyperparameter for FedGMA

To determine the optimal sign-agreement threshold $\rho$ for FedGMA, we searched the range $[0.1, 0.2, \ldots, 1.0]$, selecting the value that yields the highest average normalized accuracy in validation datasets.

### B.1.4 Hyperparameter for CCLIP

To determine the optimal threshold $\rho$ for CCLIP, for each experiment, we searched the range generated by a sequence of five evenly spaced numbers between the minimum task vector norm (including) and maximum task vector norm (excluding) used in the experiment, selecting the value that yields the highest average normalized accuracy on validation datasets.

### B.1.5 Task vector norm

In Table 7, we present the norms of the task vectors used in Section 6. Homogeneous fine-tuning task vectors, as provided by Ilharco et al. (2023), are used in Section 6.1.2, while heterogeneous fine-tuning task vectors, developed as part of our work (detailed in Appendix B.1.1), are used in Section 6.1.1.

|  | DTD | EuroSAT | GTSRB | SUN397 | SVHN | MNIST | Cars | RESISC45 |
|---|---|---|---|---|---|---|---|---|
| Homogeneous Fine-Tuning | 2.47 | 2.27 | 2.35 | 2.91 | 2.70 | 2.45 | 2.80 | 2.54 |
| Heterogeneous Fine-Tuning | 2.77 | 2.71 | 1.92 | 2.04 | 23.90 | 3.03 | 2.80 | 3.07 |

Table 7: **Norms of task vectors**

### B.2 Additional Experiments Using Task Vectors with Homogeneous Fine-Tuning

In this section, we present additional experiments on task vectors generated through a homogeneous fine-tuning process, provided by Ilharco et al. (2023). We apply Median, FedGMA, and CCLIP to these task vectors. Due to the homogeneous nature of the fine-tuning process, FedNOVA is not applicable. The hyperparameter search for each method follows the same procedure described in Appendix B.1.

Table 8 summarizes the percentage of experiments that show improvement, no change, or degradation when compared to Task Arithmetic. Additionally, Figure 3 uses histograms to depict the frequency of experiments within each range of change in average normalized accuracy.

In most cases, these Federated Learning algorithms fail to outperform Task Arithmetic. Instead, they tend to degrade the performance of the merged models, albeit usually by a small margin. These findings further reinforce the key observation discussed in Section 6.2: training heterogeneity is a critical factor in practice. Simply regulating the fine-tuning process to eliminate training heterogeneity can significantly enhance the performance of Task Arithmetic.

|  | Percentage of Improved Experiments | Percentage of Unchanged Experiments | Percentage of Degraded Experiments |
|---|---|---|---|
| Median | 5.67% | 0% | 94.33% |
| FedGMA | 9.72% | 34% | 56.28% |
| CCLIP | 49.39% | 0% | 50.61% |

Table 8: **Percentage of improved, unchanged, and degraded experiments using different methods compared to Task Arithmetic.**



Figure 3: **Histograms showing the change in average normalized accuracy for three different methods compared to Task Arithmetic by using task vectors with homogeneous fine-tuning.** The x-axis shows the difference in average normalized accuracy relative to Task Arithmetic, with positive values indicating improvement and negative values indicating degradation. The y-axis represents the number of experiments within each range of change values.

## C    Merging LLMs

### C.1    Additional Experiment Details for Merging LLMs

In this section, we present additional experimental details for merging LLMs. All experiments in this part were conducted on four NVIDIA V100 GPUs. In Table 9, we provide HuggingFace download links for fine-tuned models used in our experiments.

|  | Model | Download Link |
|---|---|---|
| Mathematical Reasoning | WizardMath-13B | https://huggingface.co/vanillaOVO/WizardMath-13B-V1.0 |
| Code Generation | Llama-2-13b-code-alpaca | https://huggingface.co/layoric/llama-2-13b-code-alpaca |
| Instruction Following | WizardLM-13B | https://huggingface.co/WizardLMTeam/WizardLM-13B-V1.2 |

Table 9: **Fine-tuned model download information**

In order to conduct hyperparameter search, we randomly split 5% of GSM8K, MATH, HumanEval and AlpacaEval into validation datasets. To determine the optimal scaling coefficient $\lambda$, we searched the range $[0.2, 0.4, 0.6, 0.8, 1.0]$ for Task Arithmetic, FedGMA and CCLIP.

To determine the optimal sign agreement threshold $\rho$ for FedGMA, notice that when there are two task vectors merged together, the sign agreement score for each coordinate is either 0 (opposite sign) or 1 (same sign). Therefore, we simply set $\rho$ to be 0.1.

To determine the optimal threshold $\rho$ for CCLIP, for each experiment, we searched the range generated by a sequence of five even spaced numbers between the minimum task vector norm (including) and maximum task vector norm (excluding) used in the experiment.

### C.2    Additional Experiment Results on Merging LLMs by Median

Table 10 presents the experimental results of applying Median to merge all three task vectors. Compared to the results in Table 5, Median enhances the merged model's mathematical reasoning capabilities by preserving most of its task vector, as its task vector has a norm of middle value. However, this improvement comes at the cost of compromising the model's ability for code generation and instruction following. These findings suggest that applying Median to task vectors generated through different fine-tuning methods may be suboptimal, highlighting the need for developing new model merging techniques.

| Tasks | Method | Mathematical Reasoning | | Code Generation | | Instruction Following |
|---|---|---|---|---|---|---|
| | | GSM8K | MATH | HumanEval | MBPP | AlpacaEval |
| Instruction Math Code | Median | 65.73 | 13.7 | 10.37 | 11.6 | 53.5 |

Table 10: **Performance of Median on merging LLMs**

## D   Generalization Ability of Task Arithmetic

In this section, we explore one potential reason behind the strong empirical performance of Task Arithmetic observed in several experiments. We conjecture this success is linked to the strong generalization capabilities that Task Arithmetic may inherit from FedAvg, or local SGD.

Research on local SGD has shown that, compared to mini-batch SGD which is the core algorithm used in standard centralized training, local SGD can offer better generalization properties (Gu et al., 2023; Lin et al., 2018; Zhu et al., 2023). Lin et al. (2018) first observed that switching to local SGD after several epochs of mini-batch SGD training enhances model generalization, leading them to propose a *post-local SGD* approach. In this scheme, local SGD is only employed in the second training phase, after initial mini-batch SGD training. This two-phase strategy mirrors Task Arithmetic, where we start with a mini-batch pre-trained model, switch to local SGD, and ultimately aggregate the updates.

Gu et al. (2023) provided theoretical insights into why local SGD improves generalization. They derived a Stochastic Differential Equation (SDE) that models the long-term behavior of local SGD, observing that it induces a larger drift term compared to standard SGD, thereby adding a regularizing effect. Later on, Zhu et al. (2023) proved that decentralized SGD is asymptotically equivalent to minimizing the loss function of an average-direction sharpness-aware minimization algorithm, which enhances generalization by seeking flatter regions in the loss landscape. This challenges the common belief that centralized training always outperforms decentralized approaches.

The similar phenomenon has been observed in the context of Task Arithmetic. For example, Yu et al. (2024) report in Table 1 that Task Arithmetic occasionally surpasses task-specific fine-tuning. In our experiments, particularly when merging LLMs, Task Arithmetic exhibits strong performance. As shown in Table 5, Task Arithmetic achieves the best results when combining three task vectors. This performance is likely attributed to the remarkable generalization ability of local SGD, even in the one-shot setting of Task Arithmetic.

