# OpenReview forum: "Task Arithmetic Through The Lens Of One-Shot Federated Learning"
_TMLR — Accepted by TMLR_

### Review · Reviewer_NweM · 2025-01-02

**Summary Of Contributions:**

Similar to another paper that also examines Task Arithmetic for multi-task learning by framing it as a one-shot Federated Learning problem, this paper takes the approach a step further ----- By leveraging well-established theoretical results from FedAvg, the authors identify two key factors that influence the performance of Task Arithmetic: data heterogeneity and training heterogeneity. Furthermore, they conducted several experiments using standard Federated Learning algorithms to enhance the effectiveness of Task Arithmetic.

**Audience:**

Yes

**Claims And Evidence:**

Yes

**Requested Changes:**

Move the Related Work to section 2, add more discussions on some related papers, and give proper credit to these works.

Clearly discuss the challenges of applying methods from FL to Task Arithmetic.

**Strengths And Weaknesses:**

After reading this submission, I have the following questions and suggestions:

1. Reorganizing Related Work and giving proper credit

I suggest the authors move the Related Work section to the second section of the paper. This would provide better context early in the discussion. Additionally, the authors should give more credit to [1], which highlights that "model merging is an extreme case of Federated Learning, where only one round of communication occurs."

Currently, the Related Work section is placed at the end of the paper, which diminishes the acknowledgment of prior contributions. As a result, it is challenging to clearly see the similarities and differences between FL and Task Arithmetic. This distinction is not well-discussed.

My impression after reading the submission is that the paper primarily applies FL methods to Task Arithmetic. However, the specific challenges and difficulties faced during this process are not clearly articulated.

2. Missing comprehensive baselines and comparisons

The paper lacks richer baselines and more detailed comparisons with [1]. This omission limits the understanding of the novelty and effectiveness of the proposed approach.
Moreover, [2], which is a well-known one-shot FL algorithm, is not discussed or compared. Including these discussions would strengthen the paper's contribution and provide more context for readers.

[1] Dataless Knowledge Fusion by Merging Weights of Language Models. ICLR 2023.

[2] DENSE: Data-Free One-Shot Federated Learning. NeurIPS 2022.

---

> ### Author Response · Authors · 2025-01-10
>
> Thank you very much for your thorough review and constructive comments on paper. We truly appreciate the time and effort you dedicated to evaluating our work. Your feedback has been incredibly valuable in helping us improve the paper.
> Below, we provide a summary of how we have addressed your comments:
>
> 1. Reorganizing Related Work: We have reorganized the section Related Work and moved it to Section 2 in order to better acknowledge previous work. Additionally, we’ve expanded the discussion on [1] and how it is related to our work.
>
> 2. Missing discussion on challenges in adapting Federated Learning algorithms: In our initial submission, we discussed challenges in adapting Federated Learning algorithms for Task Arithmetic in Appendix B. We agree that this section is very important for the comprehensiveness of the main paper. Therefore, we have moved Appendix B to the main paper Section 5.1.
>
> 3. Missing comprehensive comparisons with [1] and [2]: We have expanded the discussion on [1] in the Related Work section and on [2] in the Challenges in Adapting Federated Learning Algorithms for Task Arithmetic section. We have discussed that [1] is different from Task Arithmetic framework and [2] is not suitable for adaptation. As a result, we did not include them in the comparison. Specifically:
>
>      - The method proposed by [1], RegMean, is for merging linear layers of fine-tuned models. More importantly, in order to implement RegMean, it requires the input features of all linear layers. In contrast, the algorithms we compared only require the model weights, with the exception of FedNOVA, which also needs the number of fine-tuning iterations—information that is easily stored and accessed. In Section 2 Related Work, we have added details on how [1] is different from Task Arithmetic and our framework.
>
>     - The method in [2], DENSE, involves additional data generator training and knowledge distillation. As noted in the algorithm selection guideline in Section 5, we aim to avoid extra training when merging models. To emphasize this, we’ve cited [2] in Section 5.1 (Challenges in Adapting Federated Learning Algorithms for Task Arithmetic). With these added discussions, we hope the reasons why [1] and [2] are not included in our approach are clearer.
>
> We believe the revisions made based on your feedback have greatly improved the paper. Please let us know if there are any further questions or concerns.
>
> [1] Jin, Xisen, et al. "Dataless knowledge fusion by merging weights of language models." arXiv preprint arXiv:2212.09849 (2022).
> [2] Zhang, Jie, et al. "Dense: Data-free one-shot federated learning." Advances in Neural Information Processing Systems 35 (2022): 21414-21428.

---

### Review · Reviewer_9fnG · 2025-01-26

**Summary Of Contributions:**

In this paper, the authors formulate the Task Arithmetic (TA) problem within the Federated Learning (FL) framework as a one-shot FL problem. They extend convergence analysis from the FL domain to study the convergence properties of Task Arithmetic. The analysis highlights the impact of data and training heterogeneity on Task Arithmetic performance. Additionally, the authors adapt several FL algorithms to the Task Arithmetic setting and conduct empirical evaluations on benchmark datasets to validate their theoretical findings.

**Audience:**

Yes

**Claims And Evidence:**

Yes

**Requested Changes:**

Weakness 1: The paper would benefit from a clearer introduction to the TA problem, including formal definitions to provide a more precise understanding.

Weakness 2: The description of heterogeneity should be more precise, distinguishing between data heterogeneity, specifically statistical heterogeneity across tasks, and training heterogeneity, which refers to variations in local training steps.

Weakness 3: While the paper provides a solid theoretical analysis of the TA problem, it would be more impactful to focus on TA-specific challenges and highlight the contributions to TA directly, rather than emphasizing comparisons with FL.

**Strengths And Weaknesses:**

Strengths:
1. Robust Theoretical Analysis: The paper provides a solid and well-structured theoretical foundation.

2. Clarity and Presentation: The content is well-presented and easy to follow.

3. Comprehensive Coverage: The paper includes extensive references and thorough experiments, incorporating large language models (LLMs).

Weaknesses:
1. Unclear Problem Definition: The definition of the TA problem lacks clarity and could benefit from further elaboration.

2. Overclaims: The study focuses primarily on statistical heterogeneity across tasks and variations in training steps, yet it generalizes findings to broader aspects of data and training heterogeneity, which may be overstated.

3. Limited Novelty and Originality: The paper heavily relies on adapting existing concepts and theoretical results from FL to the TA problem, with a direct application of FL algorithms rather than proposing fundamentally new approaches.

---

> ### Author Response · Authors · 2025-02-06
>
> Thank you for the time and effort in providing detailed and constructive feedback. We are grateful for your thoughtful comments which have helped us improve the clarity of our work. Below, we provide our responses to your comments.
>
> 1. Unclear problem definition: To enhance clarity, we have added more details in the first paragraph of Section 1 Introduction to better define the problem of Task Arithmetic in the context of multi-task learning.
>
> 2.  Overclaims on data heterogeneity and training heterogeneity: We agree that this paper primarily focuses on statistical heterogeneity across tasks and variations in training hyperparameters, which we refer to data heterogeneity and training heterogeneity respectively. To clarify these terms, we have added the definition of data heterogeneity in the context of Task Arithmetic at the beginning of Section 4.1 and the definition of training heterogeneity at the beginning of Section 4.2.
>
> 3.  Limited novelty and originality: This project focused on the modest yet foundational goal of establishing that Task Arithmetic is a form of Federated Learning. This offers a novel lens through which Task Arithmetic can be analyzed, formulated, and improved using Federated Learning techniques. We have further clarified the novelty at the end of the introduction.
>
> We believe the revisions made in response to your reviews have greatly strengthened the paper. Please let us know if you have any further questions or concerns.

---

### Review · Reviewer_kFLT · 2025-02-03

**Summary Of Contributions:**

This paper establishes a novel connection between Task Arithmetic (a model merging technique) and Federated Learning, specifically one-shot Federated Averaging (FedAvg). The authors leverage theoretical insights from Federated Learning to analyze factors affecting Task Arithmetic, such as data and training heterogeneity, and propose adaptations of Federated Learning algorithms to enhance Task Arithmetic's performance.

**Audience:**

Yes

**Claims And Evidence:**

Yes

**Requested Changes:**

1. Provide an overview picture about the architecture of the paper.
2. Compare with more methods in the area of federated learning.
3. Improve the scale of the experiments and show the results.

**Strengths And Weaknesses:**

Strengths

1. The study effectively links Task Arithmetic to Federated Learning, offering a novel viewpoint on the integration of models.
2. The examination of data and training diversity under the recognized assumptions of Federated Learning is thorough and perceptive.
3. The modification of current algorithms for Task Arithmetic, along with thorough experiments, greatly enhances their practical application.
4. The experiments explore homogeneous and heterogeneous environments alike, thoroughly confirming the theoretical assertions.
5 The paper is organized logically, with clear descriptions of both theoretical and experimental elements.

Weaknesses
1. The proposed methods may face challenges when applied to extremely large-scale scenarios with highly diverse tasks.
2. The paper could benefit from a broader comparison with alternative model merging techniques beyond Task Arithmetic.
3. Lack an overview picture about the structure of the paper.

---

> ### Author Response · Authors · 2025-02-07
>
> Thank you for providing us with thoughtful comments and constructive feedback. Your suggestions have provided valuable perspectives that help us improve the clarity and impact of our paper. Below, we provide a response to each of your comment and the revision we have made.
>
> 1. Lack an overview picture about the structure of the paper: We agree that a clear overview of the paper is very important. To address this, we have revised the summary of our contributions at the end of Section 1 Introduction to provide a more structured and coherent outline of the paper.
>
> 2. Compare with more methods in Federated Learning: Due to the challenges in adapting Federated Learning algorithms for Task Arithmetic, most existing algorithms are not applicable in this scenario. Therefore, we have selected only four Federated Learning algorithms for comparison. In the initial submission, the section Challenges in Adapting Federated Learning Algorithms for Task Arithmetic was included in the Appendix. To enhance coherence and clarity, we have now moved it to the main paper as Section 5.1.
>
> 3. Improve the scale of the experiments: We acknowledge the importance of studying the performance of model merging methods in large-scale scenarios and agree that this remains a crucial research topic. This project focused on a more modest yet foundational goal: establishing that Task Arithmetic is a form of Federated Learning. Since large-scale model merging remains an open research question, we have revised Section 8 to outline our perspective on future directions, highlighting how our contribution can inform further exploration in this area. We have highlighted that in Federated Learning, there remains a significant gap in theoretical understanding and algorithmic development for addressing data and training heterogeneity in large scale experiments due to the following reasons:
>
>
>      - While existing theoretical work primarily focuses on first-order heterogeneity—such as Assumption 4.5 in our paper—recent studies [1], [2] suggest that this perspective may be insufficient to fully capture data heterogeneity in practice. This underscores the need for more refined definitions of data heterogeneity and the development of corresponding algorithms.
>
>
>      - Second, it has been observed that in practice, heterogeneity has a diminished impact on very large models, though there is no clear theoretical explanation for this phenomenon. For instance, in the model merging community, recent work [3] has shown that when merging very large models, the performance differences between various merging methods become negligible. However, the underlying reasons behind this phenomenon—specifically, how and why large model size alleviate heterogeneity—remain an open research question.
>
>     We hope that by establishing a connection between Task Arithmetic and Federated Learning, both research communities can leverage future advancements from each other, ultimately helping to address the challenges associated with scaling up.
>
> We believe these revisions based on your comments have made this paper more coherent. Please let us know if there are any further questions or concerns.
>
>
> [1] Patel, Kumar Kshitij, et al. "The limits and potentials of local sgd for distributed heterogeneous learning with intermittent communication." arXiv preprint arXiv:2405.11667 (2024).
>
> [2] Wang, Jianyu, et al. "On the unreasonable effectiveness of federated averaging with heterogeneous data." arXiv preprint arXiv:2206.04723 (2022).
>
> [3] Yadav, Prateek, et al. "What Matters for Model Merging at Scale?." arXiv preprint arXiv:2410.03617 (2024).

---

### Author Response · Authors · 2025-05-16
**Acknowledgments**

We would like to thank all the reviewers and the Action Editor for their insightful comments and constructive feedback. Their input has greatly contributed to improving the quality and clarity of our work.

---

### Decision · Action_Editor_3dod · 2025-04-21

**Recommendation:** Accept as is

**Comment:**

This paper applies task arithmetic, a model merging technique, to tackle heterogeneous challenge in federated setting. The heterogeneous is a major challenge of federated learning, thus a new solution will benefit to tackle this challenge. The proposed technique solution is technique sounds and well-supported by theoretical and experimental analysis. All three reviewers are learning to accept this paper, and there is no major concern raised.

**Audience:**

The combination of task arithmetic and federated learning is interesting. The audiences from these two domains will be interested in this work.

**Claims And Evidence:**

It claims are well supported by theoretical and experimental analysis.